# Detection of Anomalies in Data Streams Using the LSTM-CNN Model

**DOI:** 10.3390/s25051610

**Published:** 2025-03-06

**Authors:** Agnieszka Duraj, Piotr S. Szczepaniak, Artur Sadok

**Affiliations:** 1Institute of Information Technology, Lodz University of Technology, al. Politechniki 8, 93-590 Łódź, Poland; piotr.szczepaniak@p.lodz.pl; 2NESC Sp. z o.o., ul. Wojskowa 6/B1, 60-792 Poznań, Poland

**Keywords:** outliers, anomalies, deep learning methods, LSTM

## Abstract

This paper presents a comparative analysis of selected deep learning methods applied to anomaly detection in data streams. The anomaly detection results obtained on the popular Yahoo! Webscope S5 dataset are used for the computational experiments. The two commonly used and recommended models in the literature, which are the basis for this analysis, are the following: the LSTM and its more complicated variant, the LSTM autoencoder. Additionally, the usefulness of an innovative LSTM-CNN approach is evaluated. The results indicate that the LSTM-CNN approach can successfully be applied for anomaly detection in data streams as its performance compares favorably with that of the two mentioned standard models. For the performance evaluation, the $F_{1score}$ is used.

## 1. Introduction

Anomaly detection, which is one of the main data mining problems, consists of finding points or objects that do not align with the expected pattern, as related to the other points/objects in a dataset. In database research, anomalies, or exceptional patterns, are also referred to as outliers. In the literature, there is no rigid definition of what constitutes an outlier. Determining whether or not a given unusual observation is an outlier is often an intuitive process that depends strictly on the application domain. In general, an outlier can be defined as a data point (data vector) that differs significantly from the rest of the data in a dataset, or an anomalous observation that diverges from the expected pattern, which may be discovered while studying a particular phenomenon. The problem of outlier detection has been studied in the context of several domains. Not only does it provide the ability to gain new knowledge about a dataset, but it can also be used to aid the performance of machine learning, decision making, or expert systems.

Intelligent data analysis with outlier detection is currently a very rapidly developing area of research based on artificial intelligence and machine learning.

The concept of artificial intelligence was initially aimed at automating the standard human thought process. However, symbolic artificial intelligence based on defining a large number of knowledge processing rules has been extended with machine learning methods. The performance of machine learning models is driven by the exposure to large amounts of data, which allow them to learn and improve through experience—an attempt to mimic the human thought processes and behaviors employed when solving practical engineering problems.

Learning, in this context, can be described as automated search for better representations of some input data. In other words, a machine learning model aims to automatically find ways to transform data into such representations that can facilitate the performance of a particular task.

Machine learning algorithms most often use a predefined set of operations called a hypothesis space or feedback signal. There are several types of machine learning algorithms, including the following: supervised, unsupervised, reinforcement, and deep learning. It is the latter that provides the foundation for the present study. Deep learning is a subset of machine learning that employs artificial neural networks with many input, output, and hidden layers. Each layer converts the input into information that can be used by subsequent layers to perform a predictive task.

Anomaly detection problems can be solved using both supervised and unsupervised methods. An example of an unsupervised learning technique is the density-based spatial clustering of applications with noise (DBSCAN). This algorithm searches for the so-called base points and breakpoints within cluster data, treating everything outside of them as noise or unique patterns. Two other commonly used outlier detection methods are the local outlier factor (LOF) and the connectivity-based outlier factor (COF). LOF uses the nearest neighbors to detect outliers. Each data item is assigned a score indicating the degree of its deviation from its nearest neighbors. The higher the score, the higher the likelihood of the item being an outlier. The supervised anomaly detection algorithms include the One-Class SVM and the Isolation Forest. It should be noted that outlier detection encompasses a broad spectrum of techniques, making use of algorithms from different fields of computing, like intelligent classification algorithms, data grouping, decision rules, neural networks, and genetic algorithms, to name a few.

In this paper, we consider a special type of data, namely data streams. The collection, processing and rapid analysis of such data requires the application of novel methods or the modification of the existing ones. The first formal concepts and definitions of data streams in the literature appeared after 2000. In [1], the concept was formalized and data stream models were described in detail. An important point of reference in this field is the work of S. Muthukrishnan [2], which is one of the first comprehensive reviews of algorithms and techniques used for analyzing data streams. In this work, a data stream was defined as follows:

“A data stream is a sequence of data items that arrive online, are potentially unbounded in size, and must be processed under limited memory and time constraints.”

In this article, the following definition is adopted.

**Definition** **1.** 
*A data stream can be formally defined as an ordered sequence of data arriving continuously, potentially infinitely, in real time. Let S denote the data stream, then, the following holds: $S={\left\{\left(t_{i},x_{i}\right)\right\}}_{i=1}^{\infty }$*


*where*


*$t_{i}\in T$ is the timestamp (arrival time or generation time) for the i-th element.*

*$x_{i}\in X$ is the value of the i-th element (it can be a feature vector or a single value).*

*T is the time domain, and X is the data space.*



Outlier detection in data streams may be performed using statistical methods, such as sliding thresholds, the Grubbs method or (ESD), *k*-sigma, statistical hypothesis testing, Holt-Winters exponential smoothing, ARIMA, etc.

In the works of [3,4], the ARIMA method is recognized as an effective anomaly detection algorithm for time data, characterized by the so-called seasonality. Outliers are detected based on regular daily or weekly patterns. Unfortunately, there is a big problem with the dynamic determination of the seasonality period. However, Szmit et al. [5] claim that setting simple thresholds and exponential smoothing makes it possible to detect spatial anomalies. Unfortunately, the performance of statistical methods for outlier detection in data streams is not satisfactory, as these methods are sensitive to window size changes, as well as threshold changes. Data stream analysis has been performed using incremental algorithms based on hierarchical algorithms [6], or the k-means algorithm [7]. Supervised learning methods have been applied, like modified Bayesian networks, for example, as proposed in [8,9]. In addition to statistical and stochastic methods, the potential of machine learning techniques for the analysis of data streams has also been examined. Anomaly detection in data streams using deep learning methods is a relatively new area of study. Thus, there is a limited amount of literature strictly related to this topic.

The purpose of this paper is to examine the effectiveness of outlier detection using deep learning methods, with a particular emphasis on the LSTM-CNN hybrid method proposed by the authors. Webscope S5 data streams downloaded from Yahoo! were considered. To check the accuracy of outlier detection, the results of the new hybrid LSTM-CNN method were compared with those of the LSTM and LSTM autoencoder networks. The article’s contributions are as follows:The discussion of the concept of drift is carried out.Two methods for determining the anomaly detection threshold are explained, and relevant model features that enable the detection of outliers in data streams are defined.A LSTM-CNN model for outlier detection is proposed.The effectiveness of the proposed LSTM-CNN approach for detecting outliers in data streams is evaluated.In particular, a comparison of the proposed LSTM-CNN system to the already known LSTM and LSTM AE models is exhaustively performed.

The paper is organized as follows: Section 2 discusses the problem of anomaly detection in data streams. Section 3 describes and discusses the neural networks that provide the foundation for this investigation. It begins by explaining the basic concept of RNN and its extensions—the standard LSTM model and LSTM autoencoder. This is followed by a characterization of Convolutional Neural Networks (CNNs). Section 4 presents the essential part of the research. It begins by explaining the operation of the LSTM autoencoder as an outlier detector in data streams based on error reconstruction. Then, it moves on to describe in greater detail the LSTM-CNN method proposed by the authors. Section 5 is concerned with the stream datasets and methodology used for this study. Section 6 presents a discussion of the results. Finally, the conclusion gives a brief summary of the findings.

## 2. Problem Statemets

### 2.1. Anomalies in Data Streams

Any ordered set of items can be considered a data stream. The analysis of such data allows one to characterize the stream or to extract the information contained in it. The order of items under consideration is determined explicitly, by the timestamp, or implicitly, by the entrance time.

In the simplest standard form of description, the data $x_{t}$ are ordered by their sequence determined by the moment t={1,2,…,n}, while the number of moments can be finite or infinite. To provide high-quality results, the analysis of stream data requires access to a large dataset and continuous data inflow. This not only ensures consistency and reliability of results but also allows one to identify seasonality and trends.

Roughly speaking, an outlier $x_{t}^{out}\left(t=t_{out}\right)$ in a data stream is an item $x_{t}$ which differs significantly from the other items in the stream. Determining the character and significance of the anomaly is a separate problem. Outliers can be divided into the following three main types: point, contextual, and collective. However, for the analysis of data streams, a division into additive and innovative outliers is more often used. Each type can be characterized as follows:A point outlier is an object located at a significant distance from the other objects.An outlier is regarded as contextual when its occurrence depends on changes in context, such as time, weather changes, supply, demand, and others, depending on the area under consideration.A collective outlier occurs when a particular sequence or dataset deviates significantly from the dominant cardinality of the set. Individual objects in a collective outlier do not have to manifest themselves as point or context outliers simultaneously. It is believed that this type of outlier is relatively difficult to define precisely and detect.An additive outlier is recognized by observing the size of the error in finite time T, which can be caused by a simple human error, or a system error—in the case of data coming from a device. The noise in an incoming signal can also be regarded as an anomalous feature.The concept of an additive outlier was introduced by Barnet and Lewis in [10]. It is a measurement error in time *T*, 1≤T⩽N, caused by external factors, e.g., machine failure or human error.An innovative outlier is usually caused by a sudden change in a process or system. These types of outliers also include change points that evidence a change in an underlying trend.

It should be emphasized that additive outliers are treated as noise. The present study concentrates on finding sudden changes in the signal that may indicate the emergence of a new trend. Thus, it deals with innovative outliers, which can also be treated as contextual outliers.

Most of the current research concerning outlier detection in data streams emphasizes the use of the Long Short-Term Memory (LSTM) model and its variants. LSTM is preferred over the classic Recurrent Neural Network (RNN), since the latter suffers from the vanishing gradient problem. The use of recurrent neural networks for time-series analysis was proposed in [11,12,13]. Zhou et al. [14] proposed missing data reconstruction based on the LSTM network for Landsat time-series images. In [15], the authors proposed a new approach for efficient anomaly detection based on utilization of Markov and LSTM-based networks for real-time sensor data.

In [16], the performance of the LSTM autoencoder, among other methods, has been examined. The paper provides a detailed description of the model’s architecture and parameters applied. The results are evaluated using the universal F1-score. Moreover, the authors offer access to the database containing labeled outliers, which may advantageous for related research. A similar concept, also rated with the F1-score, is presented in [17]. A hybrid model combining deep learning and a support vector machine (SVM) for anomaly detection was also proposed in [18]. In [19], unsupervised hybrid anomaly detection approaches were developed. The anomaly detection was checked by hybrid models created with the combination of the Long Short-Term Memory (LSTM)-Autoencoder, the Local Outlier Factor (LOF), and the Mahalanobis distance. The LSTM-based deep stacked sequence-to-sequence autoencoder learns deep discriminative features from the time-series data by reconstructing the input data. These features are entered into the SVM and then recognized.

In [20], the LSTM autoencoder is recommended as an effective outlier detection method. The LSTM autoencoder was proposed in [21] as a model specialized for time-series data. This model learns the patterns of normal data and detects other data as outliers, utilizing the vibration signals from wind generators. In [22], the authors demonstrate the use of both a LSTM and LSTM autoencoder for solving forecasting and anomaly detection problems. A comparative study of the selected methods used for outlier detection in data streams can be found in [23]. In paper [24], the authors proposed a long short-term memory autoencoder (LSTM-AE)-based reconstruction error detector, which designs the LSTM layer in the shape of an autoencoder, to build a reconstruction error-based outlier detection model and extract latent features. In the paper by Mushtaq, et al. [25], an interesting hybrid framework is proposed. It combines a deep auto-encoder (AE) with both long short-term memory (LSTM) and bidirectional long short-term memory (Bi-LSTM) architectures. This framework is designed for intrusion detection systems. It works by first extracting optimal features using the auto-encoder and then employing LSTMs for the classification of samples into normal and anomaly categories. In [26], it has been shown that deep learning models can, under certain circumstances, outperform traditional statistical methods at forecasting. The authors proposed three deep learning architectures, namely, cascaded neural networks, reservoir computing, and long short-term memory recurrent neural networks. On the other hand, in [27], the authors proposed an anomaly detection framework for spacecraft multivariate time-series data based on temporal convolution networks (TCNs). In paper [28], a novel BI-LSTM-CNN technique for detecting malevolent traffic in smart devices was proposed. The authors combined deep learning techniques, specifically convolutional neural networks (CNN), with long short-term memory (LSTM) for the purpose of detecting and categorizing malevolent Internet traffic. In [29], the authors developed an improved training algorithm tailored for anomaly detection in unlabeled sequential data, such as time series. Their approach posits that the outputs of a meticulously designed system are derived from an unknown probability distribution, denoted as U, under normal conditions. They introduce a probability criterion grounded in the classical central limit theorem, enabling the real-time assessment of the likelihood that a given data point originates from U. This facilitates the on-the-fly labeling of the data. Real data are then fed into a deep Long Short-Term Memory (LSTM) autoencoder for training, identifying anomalies when the reconstruction error surpasses a predefined threshold. Anomaly detection has been investigated using hybrid models combining Long Short-Term Memory (LSTM) networks and Convolutional Neural Networks (CNNs). In [30], an intrusion detection model, named the CNN-LSTM, with an Attention Model (CLAM) was proposed for in-vehicle networks, particularly the Controller Area Network (CAN). The CLAM model employed one-dimensional convolutional layers (Conv1D) to effectively extract abstract features from the signal values. Similarly, in [31], a hybrid Intrusion Detection System (IDS) was developed by integrating Convolutional Neural Networks (CNNs) with Long Short-Term Memory (LSTM) networks. This model incorporated two regularization techniques—L2 regularization (L2 Reg.) and dropout—to mitigate the issue of overfitting. See also works [32,33]. The neural network methods used in this investigation were broadly described in [34,35,36,37,38].

### 2.2. The General Problem Statement

This paper seeks to address the problem of contextual anomaly detection in data streams, which is considered an important step in the development of dedicated models using a machine learning approach. A repeated occurrence of outliers or an increase in the number of anomalies in predictive models can indicate changes in the definition of the classes predicted by the model over time, the so-called concept drift. Thus, the detection of outliers in training datasets is recommended to ensure the high quality of these data.

An important aspect is the dimensionality of the problem. For multivariate datasets with all the characteristics of time series, methods dedicated to multidimensional time series can be used. One can also use models that do not take into account the temporal characteristics of the data but attempt to find point outliers in the multidimensional space. There are two main categories of outlier detection techniques as follows: supervised and unsupervised. The present study applies a supervised learning approach, as shown in Figure 1.

As illustrated below, after transferring the training data to the proposed model, the data prediction process takes place. Based on the prediction, the detection threshold is determined, above which a data point is labeled as an outlier.

There are two methods of determining the anomaly detection threshold, as follows:Detection by distance and data density. This method considers an outlier as a point. Using density-based methods, data are grouped into clusters using the distance between all cluster points in order to establish a clustering boundary. If values outside the cluster boundaries are detected, they are referred to as outliers. These methods do not apply to outlier detection in data streams.Detection by prediction. This method applies statistical models to predict the behavior of a dataset in the future. It is possible to predict the probable distribution of data and to highlight points that do not meet the assumed distribution requirements. The model is then capable of identifying all types of anomalies, in particular, collective outliers.

When creating an outlier detection model for data streams, it is important to pay attention to the following elements (model features):No changes are made to the time series concerning its time intervals.No interference with the stream length. When conducting outlier detection tests, it is necessary to pay attention to the desired properties of the anomaly detector.Analysis and predictions must be done online, so the algorithm must identify the $x_{t}$ state as normal or as an outlier before obtaining the next state $x_{t+1}$.Learning to identify successive states must be done continuously, without having to store the entire stream.The model should operate in an automated, unsupervised manner, without modifying the parameters of the algorithm during its operation.The model should automatically adjust to stream changes (its dynamics, drift).The model should detect anomalies as quickly as possible and minimize errors.

In the following sections, we examine the performance of the proposed novel hybrid neural network architecture (see Section 4.2) as compared to the other two approaches presented in Section 3.2 and Section 4.1.

## 3. Considered Basic Network Structures

### 3.1. Recurrent Neural Network

The Recurrent Neural Network (RNN) is a class of artificial neural networks that uses feedback connections. It includes a feedback loop that sends the output of processed information as an input at the next step of the sequence. A characteristic feature of the Recurrent Neural Network is its ability to memorize previously calculated sequences. Since RNNs are designed to deal with sequential data, they are well suited for time-series analysis [39]. Figure 2 illustrates the recurrent neural network architecture.

The left part of Figure 2 shows a basic RNN cell. The vector in hidden units is denoted as *h*. It follows from the laws of the architecture that the network evolves over time. There is also a feedback connecting neurons hidden in time. The symbol *t* indicates the time in which the RNN receives the current element of the sequence $x_{t}$ and the hidden state from the previous step marked as $h_{t-1}$. The next step is to update the hidden state as $h_{t}$, where the next time the network output $y_{t}$ is calculated. Thus, the current output $y_{t}$ depends on the input $x_{t}^{\prime }$ for $t^{\prime }<t$. Symbol *U* denotes the matrix of weights between the input and the hidden layers, *V* denotes the matrix of weights between transitions from one latent state to another, and *W* is a matrix the weights of the transition from hiding to exit. At each time iteration, calculations are performed according to Equations (Equation 1) and (Equation 2), where φ is the activation function and $b_{h},b_{0}$ are the biases. Function softmax is often used in the output layer when solving multiclass classification problems. The output values are represented as probabilities, and their values add up to 1.(1)$$h_{t}=\varphi \cdot \left(U\cdot x_{t}+V\cdot h_{t-1}+b_{h}\right)$$(2)$$y_{t}=softmax\left(W\cdot h_{t}+b_{0}\right)$$

There are four types of recurrent neural networks as follows [18]: one-to-one (one input and one output), one-to-many (one input and multiple outputs), many-to-one (multiple inputs and one output), and many-to-many (multiple inputs and many outputs). The disadvantages of RNNs, given for example in [40,41], include the following:These networks are usually very slow.Training can be difficult.If the activation function is used, the processing of longer sequences may overwhelm catching up in time.They suffer from the vanishing gradient problem.

Despite the above-mentioned disadvantages, RNNs are well-suited for tasks involving sequential data, in this case, data streams. These types of recurrent neural networks are trained using the Backpropagation Through Time algorithm (BPTT), which is a modification of the error backpropagation algorithm used in classical neural networks. A more detailed description of BPTT is given in [40]. The BPTT algorithm processes data sequences in the so-called time steps, using the current input data and previously received inputs. The network learns the patterns over time and the output received is used to calculate errors. Then, the network is rolled back up, and the weights are adjusted, keeping the errors in mind. The fading gradient problem may be solved using a modification of RNN called Long Short-Term Memory (LSTM).

### 3.2. Standard LSTM

The basic scheme of a LSTM cell is given in Figure 3.

For each time step *t*, three input values are considered. The first is the input value of the time series denoted as $x_{t}$, the next is the previous cell state $c_{t-1}$, and the previous hidden state $h_{t-1}$. The state of the cell serves as a memory layer that works through linear operations, where each time step information is removed or added. The learning process comprises several steps. At the beginning, a *sigmoidal* function called *forget gate layer* is used, which is responsible for checking what data from the previous output should be discarded. This dependency is defined by Equation (Equation 3), where

$f_{t}$ is the value of the gate at time t. A number between 0 and 1 specifies what information should be forgotten.φ denotes the activation function.$h_{t-1}$ represents the hidden state from the previous time step t−1.$x_{t}$ is the current input at time *t*.$b_{f}$ is the bias. This is a constant value that is added to the result of the sum of the weighted inputs.


(3)
$$f_{t}=\varphi \cdot \left(h_{t-1},x_{t}\right)+b_{f}$$


Then, it is determined whether the new data should be saved in the cell. Another layer, called the input layer, determines which values should be updated according to (Equation 4), where

$i_{t}$ is the value of the input gate at time *t*. It is a number from 0 to 1 that decides what information should be transferred to the memory location.$W_{i}$ is a matrix of weights that is trained during the training process of the LSTM network.


(4)
$$\begin{array}{c} i_{t}=\varphi \cdot \left\{W_{i}\cdot \left(h_{t-1},x_{t}\right)+b_{c}\right\} \end{array}$$


The tanh layer constructs a vector for new values that could be added to cells according to Equation (Equation 5), where

${\tilde{c}}_{t}$ means the new proposed cell state value at time *t*.tanh is the hyperbolic tangent activation function, which converts the result of the sum of the weighted inputs to a range from −1 to 1.$W_{c}$ is the weight matrix that is learned during the training process of the LSTM network.


(5)
$${\tilde{c}}_{t}=tanh\left(W_{c}\cdot \left[h_{t-1},x_{t}\right]+b_{f}\right)$$


After the transformations, we obtain (Equation 6), where the input from the cell will be written as (Equation 7) and (Equation 8).(6)$$c_{t}=f_{t}\cdot c_{t-1}+i_{t}\cdot {\tilde{c}}_{t}$$(7)$$y_{t}=\varphi \cdot \left(W_{o}\cdot \left(h_{t-1},x_{t}\right)+b_{o}\right)$$(8)$$h_{t}=y_{t}\cdot tanh\left(c_{t}\right)$$

Equation (Equation 6) represents the cell state update equation in LSTM networks, where

$c_{t}$ is a new value of the state of the memory cell at the moment *t*.$f_{t}$ is the value of the forget gate at the time *t*. This is a number between 0 and 1 that determines how much information should be forgotten from the previous state of the cell.$c_{t-1}$ indicates the previous state of the memory location at the moment t−1.$y_{t}$ means exit at time *t*.$W_{o}$ is a matrix of weights that is learned during the training process of the LSTM network.$h_{t}$ means the state hidden at time *t*.

### 3.3. LSTM Autoencoder

An autoencoder is a kind of artificial neural network that learns how to efficiently encode unlabeled data [36]. It consists of the following two parts: encoder and decoder. The first part deals with the generation and reduction of some representation of the features from the initial input *x* through the hidden layer *h*. The second part deals with the so-called reconstruction of the initial input from the output of the encoder due to the minimization of the loss function. The general purpose of an autoencoder is the conversion of high-dimensional data to low-dimensional data. We can define an input layer $\tilde{x}\in R^{l}$, the hidden layer $h\in R^{p}$, and the output layer as $x\in R_{l}$. The output layer has the same dimensionality as the input layer.

The mapping of input data is performed according to (Equation 9), where to compress the hidden representation p<l, and φ is the activation function, *W* is the weight matrix, and *b* is the bias vector. Such a representation of the hidden data can be used to reconstruct the input data after considering Equation (Equation 10), where $\tilde{\varphi },\tilde{W},\tilde{b}$ are analogous to (Equation 7) and (Equation 8). The Autoencoder architecture is shown in Figure 4.(9)h=φ·(W·x+b)(10)$$\tilde{x}=\tilde{\varphi }\cdot \left(\tilde{W}\cdot h+\tilde{b}\right)$$

The LSTM autoencoder is a combination of the autoencoder and the encoder–decoder LSTM architecture. The purpose of this model is not just to copy data from the input to the output. The autoencoder seeks to recognize the most relevant exception patterns in the training data by reducing the latent space at the input n<m [22]. It comprises the following three layers: the input layer, the hidden space, and the output layer. The training procedure consists of several steps. First, the network seeks to compress a high-dimensional data input to a low-dimensional data, which are mapped into a latent space. In the next step, the network learns to decode it back to the original input. The encoder–decoder output values are then compared with the initial data. The average error is calculated for the backpropagation reconstruction to update the weights in the network. The next step is to compare the encoder–decoder output data with the initial data. The mean reconstruction error is calculated. The weights are modified by the backpropagation method. The LSTM autoencoder is shown in Figure 5.

The input is $x\in R^{m}$, where the encoder compresses *x* to obtain the encoded representation $z=e\left(x\right)\in R^{n}$. Then, the decoder reconstructs this representation to obtain $\hat{x}=d\left(z\right)\in R^{m}$ as the output. The goal of the training process is to minimize the reconstruction error, as shown in Equation (Equation 11).(11)$$L=\frac{1}{2}\sum_{x}{∥x-\hat{x}∥}^{2}$$

### 3.4. Convolutional Neural Networks

Similar to other neural networks, a convolutional neural network, abbreviated as CNN, learns by adjusting its weights. It is important to remember that not every neuron in a given layer is connected to a neuron from the previous layer. Neurons receive a lot of input, take the weight of the sums, and pass them on to the activation function, where the output data are generated [37].

Using the appropriate filters, a CNN architecture is capable of accurately capturing the temporal and spatial details in an image. The reduction in the number of learning parameters and the reuse of weights can improve the network’s ability to achieve optimal fitting and thus improve its performance on the image dataset [38]. The CNN architecture consists of several layers. Unlike in a classic neural network, the neurons in the first layer of the CNN, called a convolutional layer, are not connected to the entire input but just to its local regions, scanning a few pixels at a time. In the second convolutional layer, there are also connections with pixels from the first layer. The low-level features obtained at the first layer are used to generate high-level features at the next layer. The artificial neurons in a convolution layer are arranged into 2D arrays, called filters, or kernels. A kernel is a small matrix of weights, which is applied to a 2D input matrix, region by region, performing an element-wise multiplication of the two matrices. The size of the kernel is significantly smaller than those of the input arrays. The main task of the second layer, the pooling layer, is to reduce the problem of overfitting. This is done by reducing the size of the feature map created in the previous layer, as a result of the multiplication process described above. The pooling operation allows one to reduce the computational cost. It is also possible to save RAM and reduce the number of parameters. In the pooling layer, the pooling process takes place, the size of which must be smaller than the size of the feature map. The size of a feature map is reduced by a factor of 2. For example, a polling layer applied to a feature map of size 8×8 (64 pixels) will result in an output feature map of 4×4 (16 pixels). The third type of layer is the fully connected layer, which comprises the last few layers in the network. This layer is fed with the output from the last pooling layer or convolutional layer. Before data are passed to the fully connected layer, they are preprocessed in the so-called flattening step, which involves transforming a three-dimensional output of a convolutional or pooling layer into a single-dimensional vector. Further processing is performed, as seen in fully connected regular neural networks. At the last stage of the CNN, an activation function is used to calculate the probability of the input belonging to a particular class softmax [42,43].

## 4. Outliers Detection Based on Deep Learning Models

### 4.1. LSTM Autoencoder as an Anomaly Detector in Data Streams Based on Error Reconstruction

According to [20], the network training process is performed in order to reconstruct the time-series data. The encoding part of the LSTM network learns a sequence of vector data, while the decoding part uses this input to reconstruct the current hidden state and the value predicted at the previous time step. The relations between the input sequence and the reconstructed sequence are shown in Figure 6.

Given *x*, $h_{E}^{\left(i\right)}$ is the hidden state of the encoder at time $t_{i}$ for i∈{1,2,…,L}, where $h_{E}^{\left(i\right)}\in R^{C}$. *C* is the number of units in the LSTM hidden layer of the encoder, the final encoder status $h_{E}^{\left(L\right)}$ serves as the initial state for the decoder. The prediction takes place on the top layer of the LSTM decoder. During the training process, the decoder takes x(i) as the input to obtain state $h_{D}^{\left(i-1\right)}$, where the prediction ${\tilde{x}}^{\left(i-1\right)}$ occurs, corresponding to $x^{\left(i-1\right)}$. During the prediction, the value ${\tilde{x}}^{\left(i\right)}$ is input to the decoder to obtain $h_{D}^{\left(i-1\right)}$ and make a prediction ${\tilde{x}}^{\left(i-1\right)}$.

Figure 6 shows examples of the iterations for three time steps L=3 over the LSTM autoencoder network. The decoder recreates the model for the sequence of step time L=3. The value of $x^{\left(i\right)}$ at time $t_{i}$ and the hidden state $h_{E}^{\left(i-1\right)}$ on encoder side w time $t_{i-1}$ are used to obtain the hidden state $h_{E}^{\left(i\right)}$ of the encoder in time $t_{i}$. The hidden state decoder, i.e., $h_{E}^{\left(3\right)}$, at the end of the input sequence serves as the initial state $h_{E}^{\left(3\right)}$ in the decoder part. To calculate ${\tilde{x}}^{\left(3\right)}=w^{T}h_{D}^{\left(3\right)}+b$, a linear layer with a matrix of weights *w* of dimensions c×m and bias $b\in R^{m}$ is used at the top of the decoder.

The LSTM autoencoder anomaly detection algorithm involves setting a detection threshold value. If a given value in the dataset exceeds the established threshold, the algorithm treats it as an exception. The mean absolute error of the MAE, i.e., the sum, is responsible for the detection of absolute errors divided by the sample size. The MAE error is defined by Equation (Equation 12), where *n* is the sample size, $y_{i}$ is the real value, and ${\tilde{y}}_{i}$ is the prediction value.(12)$$MAE=\frac{1}{n}\sum_{i=1}^{n}\left|y_{i}-{\tilde{y}}_{i}\right|$$

The value of the anomaly is determined after calculating the MAE for both the training and the test set. The anomaly is detected when the test set’s MAE error is greater than the maximum MAE error of the training set. An example of how the LSTM autoencoder algorithm works is shown in Figure 7. Outliers detected using this method have been highlighted in Figure 8.

### 4.2. A Novel LSTM-CNN Hybrid Model

In this paper, supervised learning is used because it directly leverages labeled data. This method effectively captures both the sequential dependencies handled by the LSTM and the spatial features extracted by the CNN. Unsupervised methods focus on discovering latent structures in the data (e.g., clustering, dimensionality reduction) and do not require labeled outputs. While such methods can be useful for pretraining or an exploratory analysis, they lack the capability to directly predict specific labels or outcomes, which is the main goal of the LSTM-CNN hybrid in this study.

Reinforcement Learning (RL) relies on feedback in the form of rewards and penalties, which are derived from interactions with the environment, while RL is powerful in decision-making tasks or control problems, it is computationally expensive and less suitable for static datasets where reward signals are not naturally available. The hybrid LSTM-CNN architecture is optimized for supervised tasks with well-defined input–output relationships rather than iterative environment-based learning. Approaches like Semi-Supervised and Self-Supervised Learning are beneficial in scenarios where labeled data are limited, as they leverage unlabeled data to improve performance. In this study, the availability of labeled data ensures that the supervised learning paradigm can fully exploit the architecture’s capabilities without requiring additional pretraining or assumptions about the data distribution. The preference for supervised learning in this context stems from the structured nature of the problem and the hybrid model’s strengths. However, this approach has limitations, such as dependence on labeled data, susceptibility to label noise, and potential overfitting to specific datasets. Alternative approaches, such as semi-supervised learning, could complement supervised methods in future work by reducing the reliance on labeled data or improving generalization. Supervised learning is the optimal choice for the hybrid LSTM-CNN method due to its ability to directly optimize the model for accurate predictions using labeled data. The method is tailored to extract sequential and spatial features effectively, which aligns naturally with the supervised paradigm. While alternative approaches have merits, their application would either introduce unnecessary complexity or fail to address the specific goals of this study as effectively as supervised learning.

The third architecture proposed and used in this work is a combination of two sequentially connected neural blocks—the Long Short-Term Memory (LSTM) and the Convolutional Neural Network (CNN). The hybrid LSTM-CNN method is inherently designed for tasks where the relationship between input data (e.g., time-series signals, sequential patterns, or spatial features) and the desired output (e.g., classifications, predictions, or labels) are well defined and can be explicitly modeled. Supervised learning is preferred because it directly leverages labeled data to optimize this mapping, ensuring the method effectively captures both the sequential dependencies handled by LSTM and the spatial features extracted by CNN.

The LSTM is recognized as the most effective method for time series. However, after reviewing the literature and other available neural network methods, we decided to look into ways of extending current models to improve their performance. Hence, the idea of combining the LSTM architecture with CNN.

It should be emphasized that the proposed LSTM-CNN hybrid model is analyzed in the context of sequence processing. It is well known that one-dimensional CNNs are not sensitive to the order of time steps. Of course, the stacking of multiple layers, one after another, enables the network to learn increasingly complex patterns in data sequences. However, in some cases, this may not be an effective solution. It is also worth mentioning that CNNs are much faster than regular recurrent networks. It should also be noted that LSTM networks are highly sensitive to time steps. Thus, they can be easily used to model time dependencies. As the number of steps increases, their learning process slows down significantly. After considering all these aspects, a sequential model was proposed, the diagram of which is shown in Figure 9.

In the first step, the model applies a one-dimensional convolutional layer and a pooling layer. This makes it possible to obtain a compressed representation of the input with higher-level functions. Then, this representation is passed on to the next layer, such as the LSTM layer, which is more capable of representing the sequence, preparing it for further processing. After the LSTM layer, the dropout layer is used, which allows for better regularization to reduce the risk of overtraining. The model’s construction is as follows. The input data are a three-dimensional vector (None, 5, 1), where 5 is the time step and 1 represents the time step input dimension features. At first, data are entered into the one-dimensional convolutional layer to extract features and obtain a three-dimensional output vector (None, 5, 256), where 256 is the number of filters (kernels). In the next step, the vector enters the pooling layer. This is a three-dimensional vector (None, 1, 256). The next part of the model is the input of the previously obtained vector to the LSTM layer, where the training takes place. The output data (None, 64) undergoes regularization. Here, another fully connected layer is applied to obtain the output value.

The input is a three-dimensional vector (None, 5, 1), where 5 is the time step and 1 represents the input dimensions. First, data are entered into a one-dimensional convolutional layer to extract features and produce a three-dimensional output vector (None, 5, 256). Moreover, 256 is the number of filters (kernels). In the next step, the vector enters the pooling layer, where a three-dimensional vector is obtained again (None, 1, 256). The next stage of the model is the introduction of the previously obtained vector into the LSTM layer, where training takes place, and the output data (None, 64) after the training stage go through the regularization process, where they later introduce another full connection layer to obtain the output value.

It should be emphasized that the choice of a window size of 5 time steps was guided by the nature of the data, task requirements, and a balance between computational efficiency and model performance. The window sizes were evaluated during the preliminary experiments to assess their impact on the model’s performance. A window of 5 time steps was deemed sufficient to capture relevant short-term trends and patterns in the data while avoiding the inclusion of unnecessary or redundant information. Smaller windows (e.g., 2 or 3 time steps) resulted in insufficient context for the model, leading to suboptimal feature extraction and reduced accuracy. Larger windows (e.g., 10 or 15 time steps) increased the computational cost and complexity of the model without providing significant gains in performance.

### 4.3. Training, Prediction, and Detection Processes Using the LSTM-CNN Hybrid Model

The steps of this algorithm are as follows:Input data—data used for LSTM-CNN network training.Data standardization—this process follows according to Formula (20).Network initialization—initialization of weights and biases on each LSTM-CNN layer.Calculation of the CNN layer—the input data are successively passed through the convolutional layer and the pooling layer, where features are then extracted and the output value is obtained.Calculation of the LSTM layer—the data from the previous step are calculated by the current LSTM layer and the output value is obtained.Calculation of the output layer—the fully connected layer receives data from the LSTM layers to obtain the output value.Error calculation—data from the full connection layer are compared with the real values and the error is calculated.Fulfillment of the condition—completion of a certain number of iterations, weight, and forecast error are lower than a certain threshold. If these conditions are met, the algorithm moves on to the next step. Otherwise, it goes to the error backpropagation step.Backward error propagation—propagate the calculated error backwards, update the weight and bias of each layer. Then, go back to step 4.Save the model—save the model.Input—enter data into the forecast process.Prediction—make predictions.Calculation of the prediction error—calculate the prediction error.Calculation of the exception detection threshold—based on the obtained forecasting error from the previous threshold, determine the exception detection threshold.Final result—list any exceptions found.

The block diagram of the learning process of the proposed modification is shown in Figure 10.

## 5. Datasets and Research Description

### 5.1. Stream Datasets

There are very few publicly available streaming datasets that can be used for testing outlier detection models. The present study applies the Yahoo! Webscope S5 Dataset (https://webscope.sandbox.yahoo.com/catalog.php (accessed on 24 April 2024)), which is a widely used benchmark for time-series data anomaly detection.

Yahoo! Webscope S5 is not a typical data stream in the technical sense because the data are stored in a collection. The streams are infinite and generated in real time. We process Webscope S5 step by step. Thanks to this, we simulate the appearance of subsequent records in real time. Webscope S5 data can be analyzed in the form of a stream due to the preservation of data continuity. The observations are ordered in time, which is typical for data streams. Although the data are available as complete sets, their analysis is performed on an ongoing basis. We therefore treat them as a stream simulation.

Webscope S5 consists of the folowing four classes: A1, A2, A3, and A4, each containing synthetic or real data with anomaly labels. The Yahoo! Webscope S5 data are characterized in Table 1 below.

Class A1 contains 67 datasets. These are synthetic sets that contain time series with different trends, noise, and seasonality. Real datasets, on the other hand, consist of time series that represent the aggregated values of data coming from various Yahoo! computing services. A2 is the least complex class, where each row contains about 1420 points and each step represents an hour of aggregated traffic. The series in classes A3 and A4 are longer and have about 1680 data points. They are also characterized by different trends and seasonality. In classes A2 and A3, contextual anomalies can be found, whereas class A4 contains additional trend reversal points. Class A1 is the most diverse because it contains real data. Section 6 uses four sets of this class, labeled 6, 11, 60, and 63, to examine the performance of the proposed outlier detection model.

The specific characteristics of class A1 are depicted in Figure 11, Figure 12, Figure 13 and Figure 14, showing the graphs of particular sets with the real outliers marked. Data points are marked in blue, and the outliers are marked in orange. In the further part of the study, the following notations are used: Set_1 (set of class A1 with label 6), Set_2 (set of class A1 with label 11), Set_3 (set of class A1 with label 60), and Set_4 (set of class A1 with label 63).

Only a few files from the A1 class were used for the study because it contains real data from Yahoo! services. Each of the 67 sets contains approximately 1400 data records. Table 2 shows a fragment of the dataset from class A1. The timestamp column shows the time when the data was received. The next column value specifies the values, while the last column anomaly indicates whether there is an outlier in the given value. If so, the value in that column is 1. If no outlier has been found, the value is 0.

According to the rules given in Figure 1, the algorithm proposed in the paper is based on supervised learning. Therefore, the last column anomaly is used only for the evaluation performed at the end of the study. The analysis set was divided into training and test sets in the ratio from 60% to 40%. *Z* and data standardization was performed using the scikit-learn library. The *Z* standardization process consists of normalizing a random variable, as a result of which the variable receives a mean expected value “*zero*” 0 and a standard deviation “*one*” 1. *Z* standardization is defined by Equation (Equation 13), where *x* is the non-standardized variable, μ is the mean of the training samples, and σ is the standard deviation of the training samples.(13)$$Z=\frac{x-\mu }{\sigma }$$

Normalization was performed to allow further comparison and analysis of data.

### 5.2. Research Description

Our research was carried out using the Python 3.9.12 programming language. Experiments and calculations were performed on an AMD Ryzen 5600X 6-Core 3.70 Ghz processor and 16 GB of RAM (Advanced Micro Devices, Inc., Santa Clara, CA, USA). The following libraries were used to implement the method: tensorflow, keras, sklearn, pandas, numpy, matplotlib, and plotly.

The time step was set to 5, while the shift vector was 1. To evaluate the model, the $F_{1score}$ was used. It is well suited for assessing the quality of the results because, unlike accuracy, it combines precision and recall. The $F_{1score}$ described by Formula (Equation 16) was determined based on (Equation 14) and (Equation 15), where tp means the true positive anomalies detected, fn represents the false negative anomalies detected, and fp the false positive anomalies detected.(14)$$Recall=\frac{tp}{tp+fn^{\prime }}$$(15)$$Precission=\frac{tp}{tp+fp^{\prime }}$$(16)$$F_{1score}=2\cdot \frac{Precission\cdot Recall}{Precission+Recall^{\prime }}$$

The hybrid LSTM-CNN model proposed here was compared with the LSTM network and the LSTM autoencoder, using the four datasets described in Section 5.1. For each method, the same time step, the same training and test sets, as well as the same learning parameters, namely epochs= 100, batch−size= 32, and validation−split= 0.2, were applied. Section 6 demonstrates the performance of all the methods used, illustrated with the learning graphs and the graphs of the threshold and outliers detected.

## 6. Results and Summary

### 6.1. Outlier Detection Using the LSTM Algorithm

The first stage of the research involved testing the performance of the LSTM algorithm for Set_1 (Set_1 label 6) (Figure 11). The following results were obtained:$F_{1score}=0.9106$,Number of outliers detected = 9, whereas the number of real outliers = 8.Learning time = 1.5 s.

The learning process of the LSTM network is shown in Figure 15, while Figure 16 shows the detection threshold of 1.54. The outliers detected based on the threshold determined for the LSTM network are shown in Figure 17. For Set_1, a high $F_{1score}$ was obtained.

The best results were achieved for Set_2 (Set_2, label 11). For 19 real outliers, 22 outliers were detected. The $F_{1score}$ coefficient was 0.92267, and the learning time $t_{learning}=1.7$ s.

For the third set (Set_3, label 60), the LSTM algorithm detected only 5 out of the 11 true outliers. A lower coefficient of $F_{1score}$ was also obtained as follows: 0.62929. The learning time was $t_{learning}=1.47$ s, and the detection threshold in this case was 13.3.

For the last set (Set_4, label 63), the shortest learning time $t_{learning}=1.2$ s was obtained. Unfortunately, the $F_{1score}$ was only 0.4965. The LSTM algorithm set the detection threshold at 1.59. It detected only one out of eight real outliers. It should be emphasized that Set_4 was completely different from sets Set_1, Set_2, and Set_3. The presence of various seasonalities, as well as fluctuations in the trend, seriously affected the network’s performance.

### 6.2. Outlier Detection Using the LSTM Autoencoder

In the next stage of the research, the LSTM autoencoder algorithm, abbreviated as LSTM AE, was tested. For Set_1, the detection threshold was set at 1.67. As a result, the algorithm detected 147 outliers, although the number of real outliers was only 8. In addition, the algorithm achieved the $F_{1score}$ of 0.47796. The details of the threshold determined and outliers detected are shown in Figure 18 and Figure 19, respectively. The detection threshold contributed to overstating the number of outliers. It can be concluded that the LSTM autoencoder was not capable of learning the training data, unlike the standard LSTM.

Better results were obtained for Set_2, with $F_{1score}=0.92267$. The algorithm took longer to train, with a learning time $t_{learning}=25.1\;s$. However, for the detection threshold of 0.46, the number of detected outliers was 21, whereas the number of real outliers was 19.

In the case of the third set, the detection threshold of 6.25 was set for the LSTM AE algorithm, which resulted in the detection of 5 out of 11 real outliers. The learning time $t_{learning}=13.7\;s$ and $F_{1score}=0.68382$. As shown in Figure 20, the lines of learning and validation do not converge in any way.

This means that the network was unable to learn the data patterns. However, the $F_{1score}$ is comparable to that of the previous model. The number of outliers detected by the two models is the same. For the fourth set, the $F_{1score}=0.78343$ and the learning time $t_{learning}=3.9$ s. The set threshold was approximately 1.57 and the algorithm detected six out of eight true outliers.

### 6.3. Outlier Detection Using the Hybrid LSTM-CNN Algorithm

The last stage of the research aimed to investigate the performance of the hybrid LSTM-CNN algorithm proposed in the paper. For Set_1, $F_{1score}=0.96628$ was obtained, with learning time $t_{learning}=4.9$ s. The algorithm set the detection threshold at 1.61. It detected seven out of eight real outliers. Figure 21, Figure 22 and Figure 23 illustrate the LSTM-CNN algorithm’s learning process, detection threshold, and outlier detection results, respectively.

The learning process of the LSTM-CNN algorithm is similar to that of LSTM. The lines of the learning graph converge (Figure 21). As a result, the correct detection threshold was set and a high rate of $F_{1score}$ was obtained.

For Set_2, the LSTM-CNN algorithm obtained an $F_{1score}=0.93708$, with a detection threshold equal to 0.28, and learning time $t_{learning}=3.6$ s. The algorithm detected 22 out of 10 outliers.

Due to its diversity, Set_3 represented the biggest challenge for the hybrid method. The algorithm obtained a $F_{1score}=0.73198$, with a detection threshold =6.9, and learning time $t_{learning}=5.3$ s. Only 6 out of 11 true outliers were detected. As can be seen in Figure 24, at some point, the training and validation curves began to move slightly closer to each other, which is representative of some progress in the network’s learning performance. The $F_{1score}$ score reached a satisfactory level.

With a threshold of 6.9, determined for Set_3 (Figure 25), only six outliers were detected (Figure 26).

For Set_4, shown in Figure 14, the hybrid model’s learning performance is somewhat similar to that obtained for Set_3. The following results were obtained:$F_{1score}=0.88347$.Learning time $t_{learning}=1.7$ s.Threshold = 1.56.Number of outliers detected = 5, whereas the number of real outliers = 8.

A summary of the results obtained by the LSTM, LSTM AE, and LSTM-CNN algorithms for all the sets analyzed is given in Table 3. The number of outliers detected by each algorithm in all the sets is given in Table 4.

## 7. Conclusions

This study has investigated the performance of a hybrid LSTM-CNN approach for outlier detection in data streams. It was shown that the proposed method outperforms the other two models—LSTM and LSTM AE. The $F_{1score}$, which was used as a performance indicator, was highest for the hybrid approach. For Set_ 1, the LSTM-CNN model outperforms the LSTM AE method, with the $F_{1score}$ being higher by as much as 0.49. The value obtained by the LSTM model is only 0.06 lower, which is still a satisfactory result. For Set_2, which turned out to be the easiest to learn, the highest value of $F_{1score}=0.94$ was obtained. It can be seen from Table 3 that LSTM and LSTM AE achieved the same result $F_{1score}=0.92$, whereas the value obtained by the LSTM-CNN model was higher by 0.02. For Set_3, LSTM-CNN also outperformed the other methods, with an $F_{1score}$ higher than LSTM and LSTM AE by 0.10 and 0.05, respectively. The same results were achieved for Set_4. Significant differences were observed in terms of the learning time. The LSTM model, the simplest in construction, was the fastest to learn. The time taken by the algorithm to train on each set did not exceed two seconds. The reconstruction-based LSTM AE was the slowest to learn. The LSTM-CNN model demonstrated a slightly longer learning time as compared to the LSTM algorithm. However, in this case, it should be noted that the hybrid LSTM-CNN model has more layers than the LSTM. In addition, the difference in learning time between the LSTM and LSTM-CNN is very small and does not significantly affect the outlier detection results.

When it comes to the learning performance, LSTM AE produced the least satisfactory results. For Set_2, the training and validation lines were closest to each other and the MSE measurement error was the smallest. In the remaining sets, the LSTM AE model failed to learn the time-series patterns. This is best illustrated by the threshold and outliers graphs, where the number of detected outliers far exceeds the number of real outliers. This is because insufficient learning has lowered the detection threshold. As indicated by the line plots of training and validation loss, the networks LSTM and LSTM-CNN demonstrated a similar learning performance, the new model performing better, as it reached a higher $F_{1score}$. The novel LSTM-CNN method was able to detect 7 out of 8, 22 out of 19, 6 out of 11, and 5 out of 8 real outliers in Set_1, Set_2, Set_3, and Set_4, respectively. Thus, it may be concluded that the hybrid approach is well suited for outlier detection problems in data streams.

As for the memory requirements of the proposed solution, the LSTM autoencoder primarily consumes memory for network weights and hidden states, typically ranging from 500 kB to several MB. The LSTM-CNN introduces a CNN layer, which increases parameter count but reduces the number of input features to the LSTM. This may lead to lower overall memory consumption depending on the complexity of the input data. Both architectures fall within the same order of magnitude in memory consumption, but LSTM-CNN has the advantage of reducing input dimensionality, which can improve efficiency. Simple statistical techniques such as PCA and regression require significantly less memory, typically in the range of tens to hundreds of kB, as they do not rely on large trainable weight matrices or hidden states. In contrast, both LSTM autoencoders and LSTM-CNN models require hundreds of kB to several MB, making them considerably more memory-intensive. Compared to the LSTM autoencoder, the proposed LSTM-CNN architecture introduces additional parameters from the CNN layer but may reduce overall memory consumption by decreasing the number of input features to the LSTM. While both architectures require significantly more memory than simple statistical methods, the CNN component helps optimize memory usage by reducing redundant input dimensions before they reach the LSTM layer.

In this article, we propose the combination of LSTM and CNN to leverage their complementary strengths for outlier detection in sequential and high-dimensional data. Each individual method has certain limitations that are addressed when used together. LSTMs excel at capturing long-term dependencies and temporal patterns in sequential data, making them ideal for time-series analysis, but they struggle with efficiently modeling spatial or local patterns in high-dimensional data, as their design focuses primarily on sequential dependencies. Furthermore, LSTMs are computationally intensive and prone to overfitting when applied to complex datasets with a high variability. Convolutional Neural Networks are highly effective at extracting local patterns and spatial features due to their convolutional structure, which operates on subsets of the input data. Unfortunately CNNs are less suited for handling sequential dependencies or temporal dynamics because they are primarily designed for fixed-size inputs with a focus on local correlations. The hybrid LSTM-CNN model combines the strengths of both architectures to address their individual limitations as follows: CNNs extract features from the data, which are then processed by LSTMs to capture temporal dependencies. The CNN layer reduces the dimensionality of the input data, feeding only the most relevant features into the LSTM. This reduces overfitting and improves the robustness of the model, especially when dealing with high-dimensional datasets. By combining these methods, the computational inefficiencies of LSTMs and the sequence-insensitivity of CNNs are mitigated. The hybrid approach ensures that the strengths of one architecture compensate for the weaknesses of the other. The combination of the LSTM and CNN is particularly important for outlier detection.

The model proposed in this article has several limitations. The hybrid LSTM-CNN method was applied to labeled data, which may restrict its applicability in scenarios where annotated datasets are scarce or unavailable. The computational demands of the hybrid LSTM-CNN model, while manageable for medium-sized datasets, could become a challenge when working with large-scale datasets, particularly in the context of big data. The model’s sensitivity to hyperparameter selection poses another limitation. Achieving optimal performance may require domain-specific expertise and significant effort in fine-tuning. Despite these limitations, the proposed method demonstrates significant potential in various real-world applications, particularly in anomaly detection systems. For example, for detecting faults or irregularities in industrial systems and IoT sensor networks; for identifying abnormal patterns in physiological signals, such as ECG or EEG data; and for detecting unusual trends or anomalies in financial time-series data, such as stock market trends. These examples illustrate the versatility of the hybrid LSTM-CNN approach in addressing complex, real-world problems involving sequential and high-dimensional data. The proposed comparative analysis of anomaly detection methods based on streaming data can be directly applied in the context of intelligent monitoring systems, IoT sensors, and temporal data analysis. The importance of anomaly detection for sensor systems is of great importance. Modern sensor systems generate huge amounts of streaming data, which require effective methods for analysis and real-time anomaly detection so the models presented in the article (LSTM, LSTM autoencoder, and LSTM-CNN) may be of interest to readers involved in sensor signal processing.

## Figures and Tables

**Figure 1 sensors-25-01610-f001:**
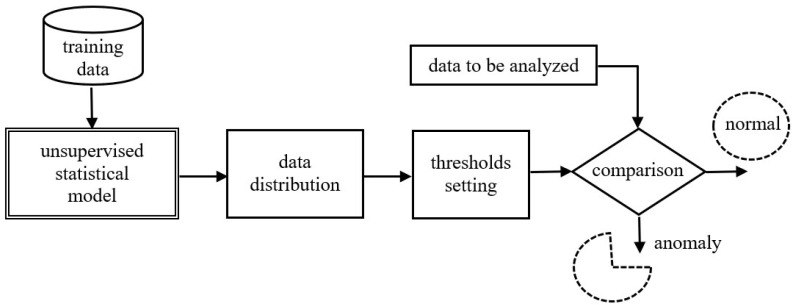
Schema of the supervised anomaly detection approach.

**Figure 2 sensors-25-01610-f002:**
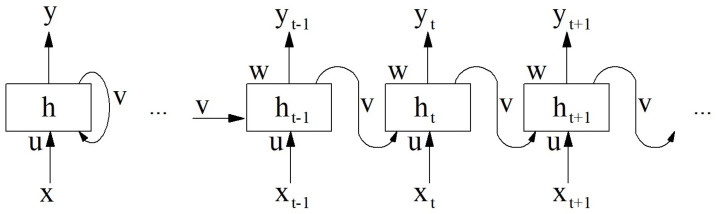
RNN network architecture diagram.

**Figure 3 sensors-25-01610-f003:**
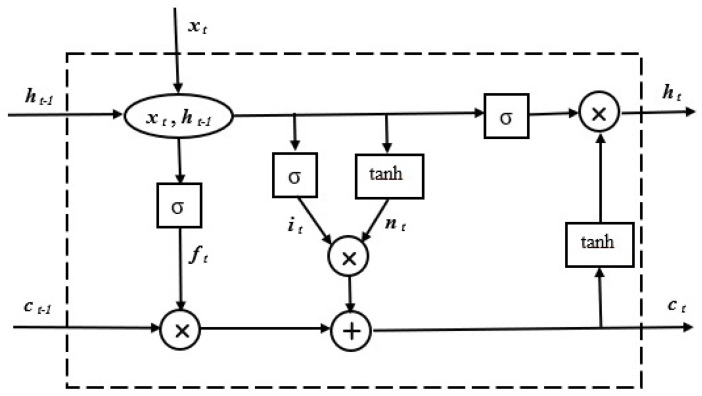
LSTM network cell.

**Figure 4 sensors-25-01610-f004:**
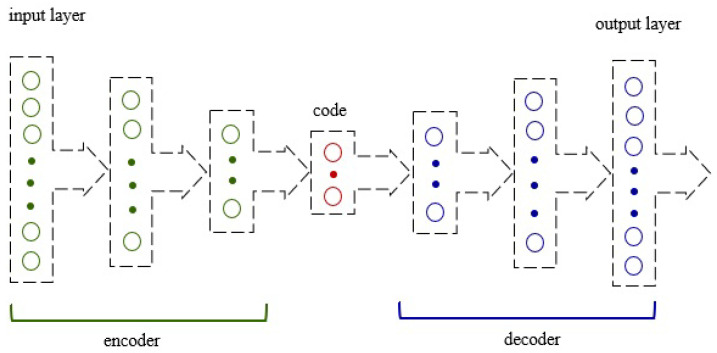
The autoencoder architecture.

**Figure 5 sensors-25-01610-f005:**
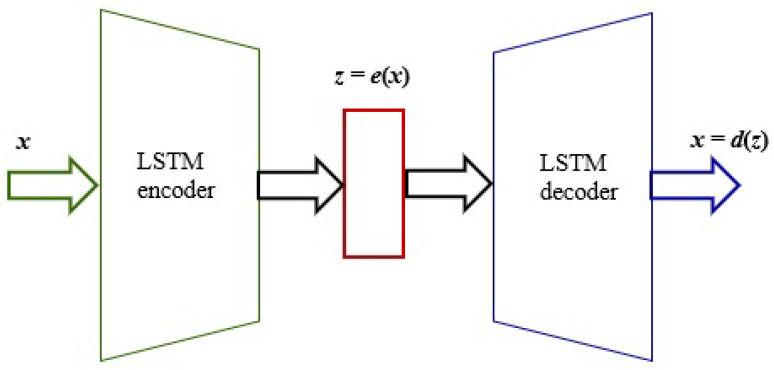
The LSTM autoencoder architecture.

**Figure 6 sensors-25-01610-f006:**
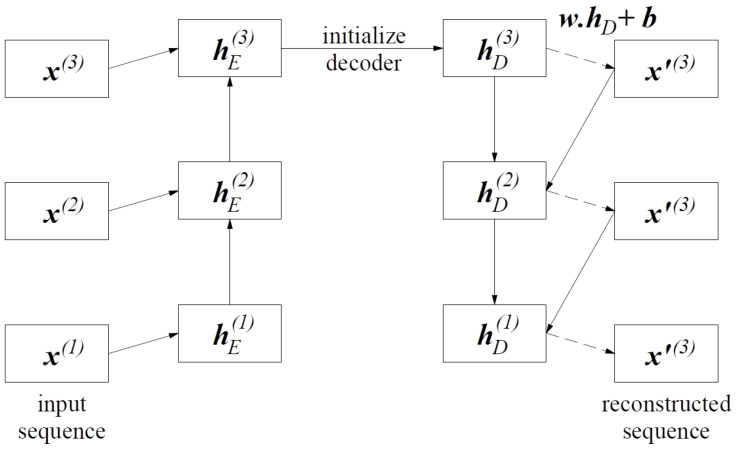
Sequence of items in the encoder–decoder network.

**Figure 7 sensors-25-01610-f007:**
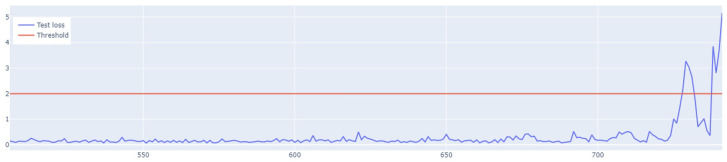
Graph of the line of the designated outlier detection threshold. The blue line represents the values, whereas the red line illustrates the outliers.

**Figure 8 sensors-25-01610-f008:**
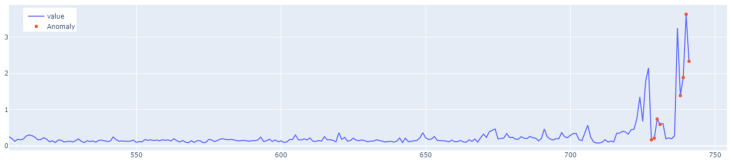
Graph with the highlighted outliers detected by LSTM autoencoder. The blue line represents the values, whereas red denotes the outliers.

**Figure 9 sensors-25-01610-f009:**
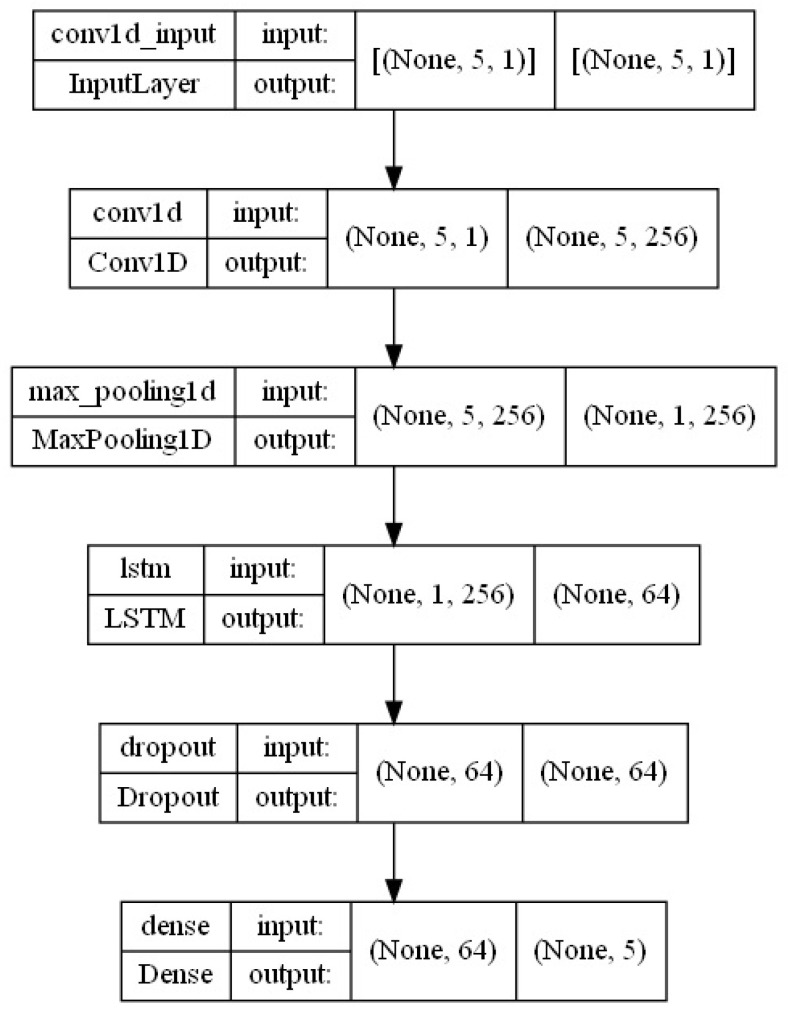
LSTM-CNN network diagram.

**Figure 10 sensors-25-01610-f010:**
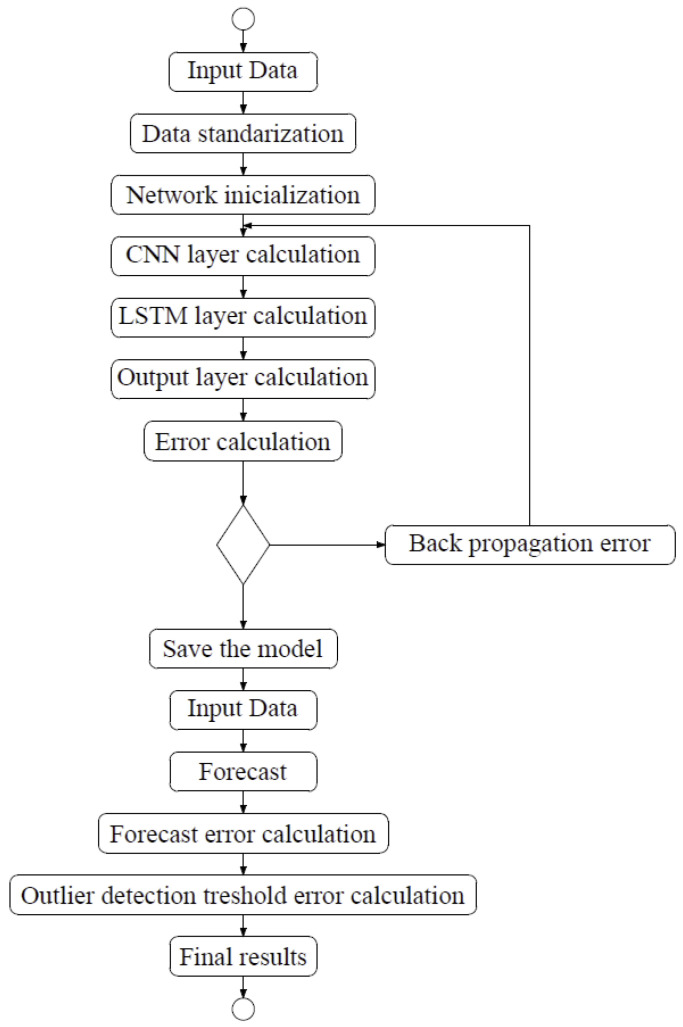
Learning, prediction, and outlier detection using the LSTM-CNN algorithm.

**Figure 11 sensors-25-01610-f011:**
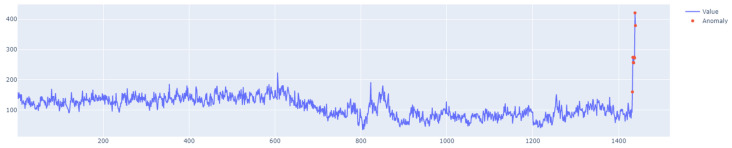
A1 specification. Set_1 with label 6.

**Figure 12 sensors-25-01610-f012:**
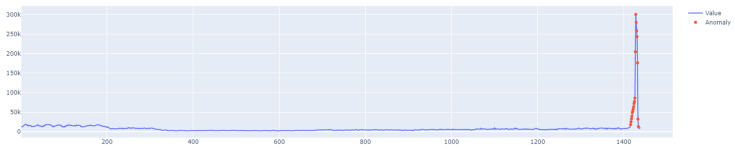
A1 specification. Set_2 with label 11.

**Figure 13 sensors-25-01610-f013:**
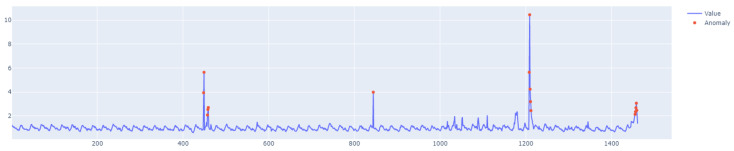
A1 specification. Set_3 with label 60.

**Figure 14 sensors-25-01610-f014:**
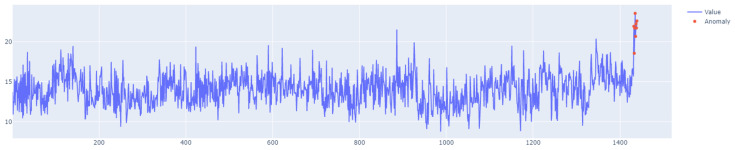
A1 specification. Set_4 with label 63.

**Figure 15 sensors-25-01610-f015:**
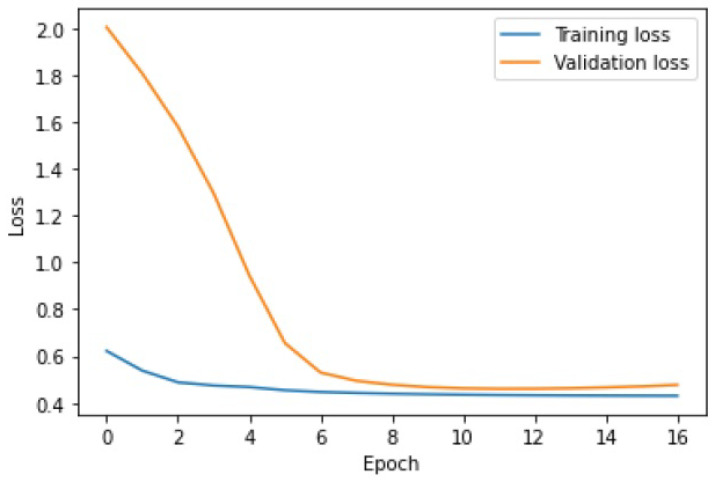
Graph of the LSTM algorithm’s learning performance for Set_1 (label 6).

**Figure 16 sensors-25-01610-f016:**
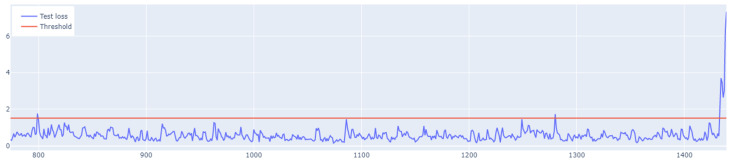
Graph of the LSTM algorithm’s detection threshold for Set_1 (label 6).

**Figure 17 sensors-25-01610-f017:**
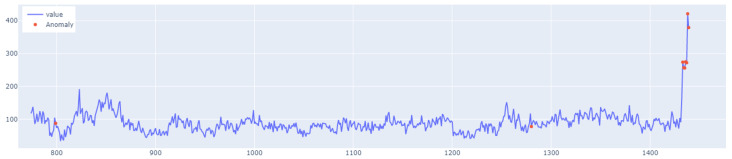
Graph of outliers detected by the LSTM algorithm for Set_1 (label 6).

**Figure 18 sensors-25-01610-f018:**
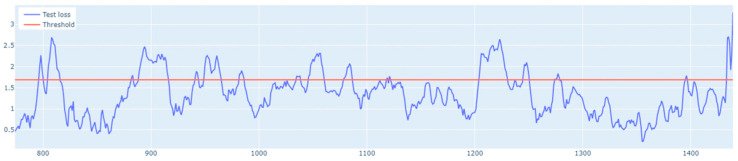
Detection threshold of the LSTM AE algorithm for Set_1.

**Figure 19 sensors-25-01610-f019:**
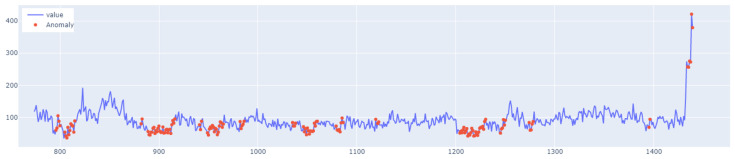
Outliers detected by LSTM AE for Set_1.

**Figure 20 sensors-25-01610-f020:**
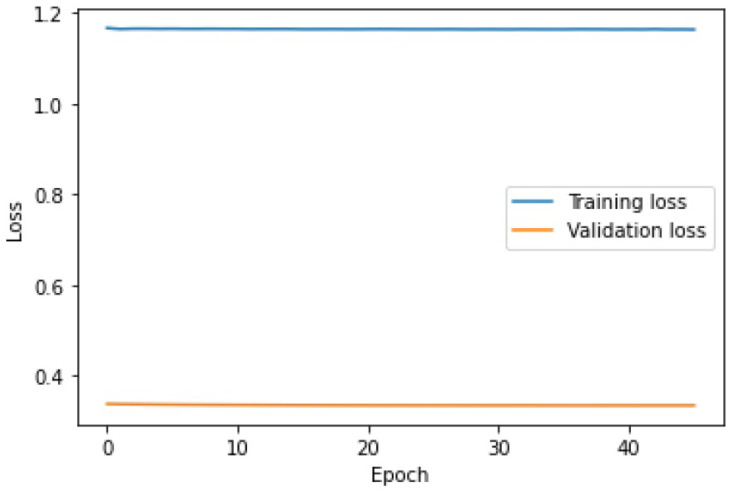
Learning process of LSTM AE for Set_1.

**Figure 21 sensors-25-01610-f021:**
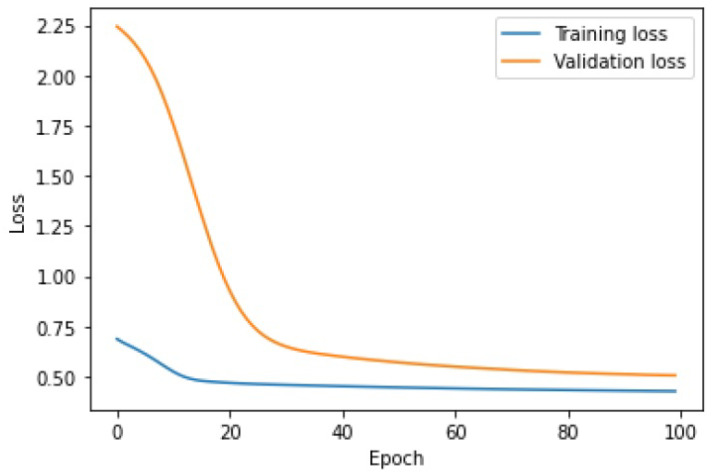
Graph of the LSTM-CNN algorithm’s learning performance for Set_1.

**Figure 22 sensors-25-01610-f022:**
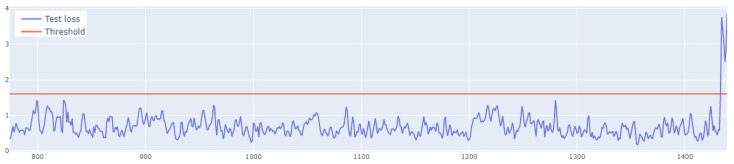
Graph of the LSTM-CNN algorithm’s detection threshold for Set_1.

**Figure 23 sensors-25-01610-f023:**
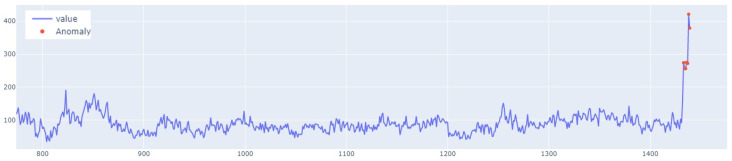
Graph of outliers detected using the LSTM-CNN algorithm for Set_1.

**Figure 24 sensors-25-01610-f024:**
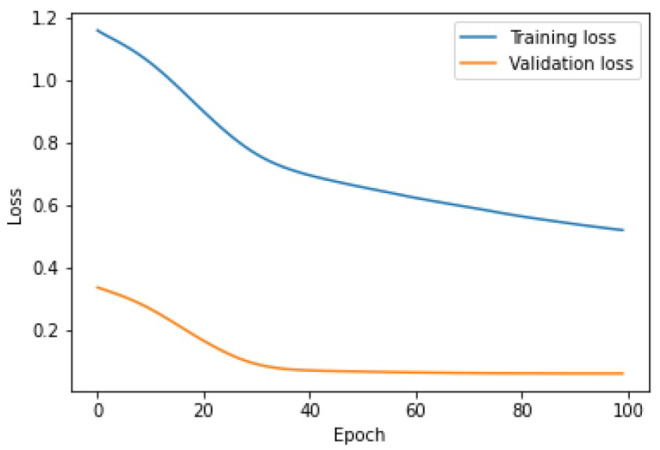
Graph of the learning performance of LSTM-CNN for Set_3.

**Figure 25 sensors-25-01610-f025:**
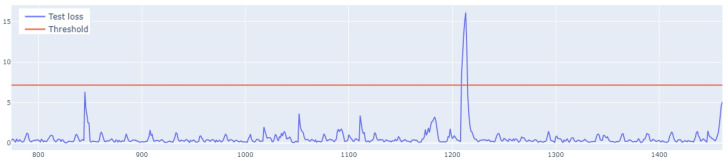
Graph of the detection threshold set using the LSTM-CNN algorithm for Set_3.

**Figure 26 sensors-25-01610-f026:**
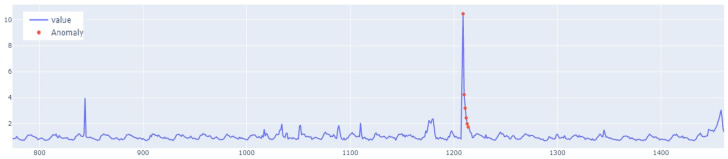
Graph of outliers detected using the LSTM-CNN algorithm for Set_3.

**Table 1 sensors-25-01610-t001:** Characteristics of Yahoo! file classes.

Data Class	Number of Dataset	Outliers Type	Number of Outliers	Outliers to the Data
A1—real	67	Contextual or complex	1669	0.0176
A2—synthetic	100	Contextual	466	0.033
A3—synthetic	100	Contextual	466	0.0056
A4—synthetic	100	Contextual	1045	0.0062

**Table 2 sensors-25-01610-t002:** Class A1 dataset structure.

*Timestamp*	*Value*	*Anomaly*
0	0.00	0
1	0.091758	0
2	0.172297	0
…	…	…
1417	0.197441	0
1418	0.161966	0
1419	0.111648	0

**Table 3 sensors-25-01610-t003:** Values of the $F_{1score}$ and learning time $t_{learning}$ results obtained by the three algorithms for all four sets.

	LSTM		LSTM AE		LSTM-CNN	
	f1_Score	Trening	f1_Score	Trening	f1_Score	Trening
Set_1	0.91	1.5 s	0.48	4.8 s	0.97	4.9 s
Set_2	0.92	1.7 s	0.92	25.1 s	0.94	3.6 s
Set_3	0.63	1.4 s	0.68	13.7 s	0.73	5.4 s
Set_4	0.5	1.2 s	0.78	3.9 s	0.88	1.7 s
mean	0.74		0.75		0.88	

**Table 4 sensors-25-01610-t004:** Number of outliers detected by the LSTM, LSTM AE, and LSTM-CNN algorithms across the four sets.

Name of Dataset	Number of Real Outliers	Number of Outliers Detected		
		LSTM	LSTM AE	LSTM-CNN
Set_1	8	9	147	7
Set_2	19	22	21	22
Set_3	11	5	5	6
Set_4	8	1	6	5

## Data Availability

Data are contained within the article.

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
