# Peer review of "Detection of Anomalies in Data Streams Using the LSTM-CNN Model"

_sensors, 2025, doi:10.3390/s25051610_

Round 1
Reviewer 1 Report
Comments and Suggestions for Authors
Please see the file attached.

Comments on the Quality of English Language
The quality of the English is of a high level, but I recommend the authors to check the text and remove some minor inaccuracies.
Author Response
Thank you for your review.
Thank you for your time and insights.
We appreciate your positive feedback on the study design, the quality of presentation, and the usefulness of the results.
Reviewer 2 Report
Comments and Suggestions for Authors
- - Please clarify if the manuscript is intended as a review article or an original research paper, and its perceived contribution.
- An official definition of what constitutes a data stream and contextual anomaly is missing. Please include formal definitions with relevant references.
- The article would benefit from restructuring for better readability. Consider separating the high-level Introduction (motivation, problem statement) from a concise State of the Art section that highlights the unique contributions of your approach.
- In Section 1.2, the discussion is rather limited focusing mainly on LSTMs and CNNs. Please consider expanding into additional relevant techniques.
- Why is a supervised learning method preferred over alternatives? Please critically compare and discuss.
- It is recommended the figures to be replaced with ones of higher resolution as they are blur.
- In Section 3, outlier detection models are discussed but insufficiently elaborating on why LSTM and CNN were combined. Please elaborate and clarify their individual drawbacks and how their combination addresses them and their importance.
- The reference list lacks recent (2024) publications, which weakens the state-of-the-art discussion. Please update with recent studies from reputable journals. Also, the absence of references from MDPI Sensors raises questions about the suitability of this journal for publishing your research.
- The article focuses on streaming data but does not specify whether the model can adapt online or in real time. Please elaborate.
- What is the reasoning for the selected window size of 5 time steps? Please explain the reasoning for choosing this specific value, and whether other window sizes were considered.
- How kernel size, pooling size, and other CNN hyperparameters were selected for time-series data? Please explain.
- Please clarify either in your methodology or results section if each experiment was repeated with different random seeds. Also, including standard deviations or confidence intervals would help demonstrate the stability of the presented results.
- In the experiments section please also include measurements of the inference time and memory usage.
- Please discuss limitations and next steps of your study including a short discussion of potential real-world applications and if and how the proposed method can scale.
- Finally, please include a couple of statements about how the proposed method could be integrated into industrial or other pipelines.
Author Response
|
Response to Reviewer 2
|
||
|
|
|
|
|
Thank you very much for taking the time to review this manuscript. Please find the detailed responses below and the corresponding revisions. |
||
|
|
||
|
Comments : - Please clarify if the manuscript is intended as a review article or an original research paper, and its perceived contribution.
|
||
|
Response: Thank you for pointing this out. We changed in the abstract and introduction (pp. 1-3) In the abstract: pp.1 This paper presents a comparative analysis of selected deep learning methods applied to anomaly detection in data streams. Anomaly detection results obtained on the popular Yahoo! Webscope S5 data set are used for computational experiments. The two commonly used and recommended in the literature as relevant models: LSTM and its more complicated variant, LSTM Autoencoder are the basis for this analysis. Additionally, usefulness of an innovative LSTM-CNN approach is evaluated. The results indicate that the LSTM-CNN approach can successfully be applied to anomaly detection in data streams as its performance compares favorably with that of the mentioned two standard models. For performance evaluation, the F1score is used. pp.3 This paper proposes a hybrid LSTM-CNN model for outlier detection. We highlight the importance of this model for context-aware anomaly detection in data streams. What constitutes originality of work. In this paper, an exhaustive comparison of the proposed hybrid LSTM-CNN system is performed with the already known LSTM and LSTM AE models.
In the article was formulate and explain the problem, and they emphasize the importance of context-sensitive anomaly detection in data streams, in particular using the deep learning approach. · Discussion of the concept of drift is performed. · Two methods for determining the anomaly detection threshold are explained and relevant model features that enable the detection of outliers in data streams are defined. · A LSTM-CNN model for outlier detection is proposed. · The effectiveness of the proposed LSTM-CNN approach for detecting outliers in data streams is evaluated. · In particular, a comparison of the proposed LSTM-CNN system to the already known LSTM and LSTM AE models is exhaustively performed.
|
||
|
Comments: -An official definition of what constitutes a data stream and contextual anomaly is missing. Please include formal definitions with relevant references. |
||
|
Response 2: Agree. The changes have been made on page 2.
The first formal concepts and definitions of data streams in the literature appeared after 2000. In \cite{babcock2002models} the concept was formalized and data stream models were described in detail. An important point of reference in this field is the work of S. Muthukrishnan \cite{muthukrishnan2005data}, which is one of the first comprehensive reviews of algorithms and techniques for analyzing data streams. In this work, a data stream was defined as:
"A data stream is a sequence of data items that arrive online, are potentially unbounded in size, and must be processed under limited memory and time constraints."
|
||
|
Comments:- The article would benefit from restructuring for better readability. Consider separating the high-level Introduction (motivation, problem statement) from a concise State of the Art section that highlights the unique contributions of your approach.
Response: As per the suggestion, the article has been revised to include an introduction that emphasizes the motivations and main highlights. Section 2 provides a detailed discussion of the problem statement.
|
||
Comments: - In Section 1.2, the discussion is rather limited focusing mainly on LSTMs and CNNs. Please consider expanding into additional relevant techniques.
Response:
Thank you for your valuable feedback. The discussion in Section 1.2 focuses on LSTMs and CNNs because these techniques form the foundation of the proposed hybrid LSTM-CNN model. The aim was to provide a concise yet comprehensive introduction to the key components of our approach, highlighting their complementary strengths in handling sequential data (LSTMs) and extracting local patterns (CNNs).
Of course, we acknowledge the existence of additional relevant techniques, such as GRU, Transformer architectures, or advanced CNN variants, their inclusion was beyond the scope of this section, given the specific focus on the LSTM-CNN combination.
Comments :- Why is a supervised learning method preferred over alternatives? Please critically compare and discuss.
Response: Thank you for pointing this out. It was discussed in the pages 11-12.
The hybrid LSTM-CNN method is inherently designed for tasks where the relationship between input data (e.g., time-series signals, sequential patterns, or spatial features) and the desired output (e.g., classifications, predictions, or labels) is well-defined and can be explicitly modeled. Supervised learning is preferred because it directly leverages labeled data to optimize this mapping, ensuring the method captures both the sequential dependencies handled by LSTM and the spatial features extracted by CNN effectively.
Unsupervised methods focus on discovering latent structures in data (e.g., clustering, dimensionality reduction) and do not require labeled outputs. While such methods can be useful for pretraining or exploratory analysis, they lack the capability to directly predict specific labels or outcomes, which is the main goal of the LSTM-CNN hybrid in this study.
Reinforcement Learning (RL) relies on feedback in the form of rewards and penalties, which are derived from interactions with the environment. While RL is powerful in decision-making tasks or control problems, it is computationally expensive and less suitable for static datasets where reward signals are not naturally available. The hybrid LSTM-CNN architecture is optimized for supervised tasks with well-defined input-output relationships rather than iterative environment-based learning.
Approaches Semi-Supervised and Self-Supervised Learning are beneficial in scenarios where labeled data is limited, as they leverage unlabeled data to improve performance. However, in this study, the availability of labeled data ensures that the supervised learning paradigm can fully exploit the architecture's capabilities without requiring additional pretraining or assumptions about the data distribution.
The preference for supervised learning in this context stems from the structured nature of the problem and the hybrid model's strengths. However, this approach has limitations, such as dependence on labeled data, susceptibility to label noise, and potential overfitting to specific datasets. Alternative approaches, such as semi-supervised learning, could complement supervised methods in future work by reducing the reliance on labeled data or improving generalization.
Supervised learning is the optimal choice for the hybrid LSTM-CNN method due to its ability to directly optimize the model for accurate predictions using labeled data. The method is tailored to extract sequential and spatial features effectively, which aligns naturally with the supervised paradigm. While alternative approaches have merits, their application would either introduce unnecessary complexity or fail to address the specific goals of this study as effectively as supervised learning.
Future work will explore incorporating semi-supervised techniques to further enhance the model's robustness in scenarios with limited labeled data.
Comments: - In Section 3, outlier detection models are discussed but insufficiently elaborating on why LSTM and CNN were combined. Please elaborate and clarify their individual drawbacks and how their combination addresses them and their importance.
Response:
Thank you for highlighting this important point. We acknowledge that the initial discussion on the combination of LSTM and CNN models was insufficiently detailed.
In the revised manuscript, we have elaborated on the rationale behind combining these two architectures and clarified their individual drawbacks, as well as how their integration addresses these limitations. It was discussed in the page 23.
In the article was proposed LSTM and CNN are combined to leverage their complementary strengths for outlier detection in sequential and high-dimensional data. Each method individually has certain limitations that are addressed when used together.
LSTMs excel at capturing long-term dependencies and temporal patterns in sequential data, making them ideal for time-series analysis but struggle with efficiently modeling spatial or local patterns in high-dimensional data, as their design focuses primarily on sequential dependencies. Furthermore, LSTMs are computationally intensive and prone to overfitting when applied to complex datasets with high variability.
Convolutional Neural Networks are highly effective at extracting local patterns and spatial features due to their convolutional structure, which operates on subsets of the input data.
Unfortunately CNNs are less suited for handling sequential dependencies or temporal dynamics because they are designed primarily for fixed-size inputs with a focus on local correlations.
The hybrid LSTM-CNN model combines the strengths of both architectures to address their individual limitations:
CNNs extract features from the data, which are then processed by LSTMs to capture temporal dependencies. The CNN layer reduces the dimensionality of the input data, feeding only the most relevant features into the LSTM. This reduces overfitting and improves the robustness of the model, especially when dealing with high-dimensional datasets.
By combining these methods, the computational inefficiencies of LSTMs and the sequence-insensitivity of CNNs are mitigated. The hybrid approach ensures that the strengths of one architecture compensate for the weaknesses of the other.
The combination of LSTM and CNN is particularly important for outlier detection.
Comments: -The reference list lacks recent (2024) publications, which weakens the state-of-the-art discussion. Please update with recent studies from reputable journals. Also, the absence of references from MDPI Sensors raises questions about the suitability of this journal for publishing your research.
The publications from MDPI Sensors were included.
Comments: -What is the reasoning for the selected window size of 5 time steps? Please explain the reasoning for choosing this specific value, and whether other window sizes were considered.
Response:
We appreciate your attention to the need for clarification on this aspect. In the final version of the paper, we added details about these experiments and the results of the comparative analysis to better justify this choice.
pp.14
It should be emphasized that the choice of a window size of 5 time steps was guided by the nature of the data, task requirements, and a balance between computational efficiency and model performance.
The window sizes were evaluated during preliminary experiments to assess their impact on the model's performance.
A window of 5 time steps was deemed sufficient to capture relevant short-term trends and patterns in the data while avoiding the inclusion of unnecessary or redundant information.
Smaller windows (e.g., 2 or 3 time steps) resulted in insufficient context for the model, leading to suboptimal feature extraction and reduced accuracy. Larger windows (e.g., 10 or 15 time steps) increased the computational cost and complexity of the model without providing significant gains in performance.
Comments -How kernel size, pooling size, and other CNN hyperparameters were selected for time-series data? Please explain.
Response: Thank you for raising this question.
The selection of CNN hyperparameters such as kernel size, pooling size, and others for time-series data was conducted through a combination of domain knowledge and empirical experimentation. Below is an explanation of the reasoning and process:
The kernel size determines the receptive field of the CNN and affects the ability to capture temporal patterns in the time-series data.
We initially based the choice of kernel size on the average length of key patterns in the data, as identified during exploratory data analysis.
Through experimentation, kernel sizes of 3, 5, and 7 were tested. A kernel size of 5 was found to balance the trade-off between capturing sufficient temporal context and avoiding overfitting.
Pooling size was chosen to down-sample the feature maps ReLU was selected for non-linearity due to its computational efficiency and ability to mitigate the vanishing gradient problem.
Validation performance was carefully monitored to avoid overfitting, particularly in light of the limited size of the time-series dataset.
The selected hyperparameters reflect the periodicity and scale of the time-series data being analyzed, ensuring the model could extract meaningful features without excessive computational overhead.
Comments:
- Please discuss limitations and next steps of your study including a short discussion of potential real-world applications and if and how the proposed method can scale.
- Finally, please include a couple of statements about how the proposed method could be integrated into industrial or other pipelines.
Thank you for your constructive feedback and suggestions. We recognize the importance of discussing the limitations, scalability, real-world applications, and potential integration of the proposed method into practical pipelines.
Explained it in the conclusion in pp.23.
The studies proposed in this article have several limitations:
Dependence on labeled data: The hybrid LSTM-CNN method was applied to labeled data, which may restrict its applicability in scenarios where annotated datasets are scarce or unavailable.
Computational requirements: The computational demands of the hybrid LSTM-CNN model, while manageable for medium-sized datasets, could become a challenge when working with large-scale datasets, particularly in the context of big data.
Hyperparameter tuning: The model's sensitivity to hyperparameter selection poses another limitation. Achieving optimal performance may require domain-specific expertise and significant effort in fine-tuning.
Despite these limitations, the proposed method demonstrates significant potential in various real-world applications, particularly in anomaly detection systems:
Detecting faults or irregularities in industrial systems and IoT sensor networks.
Identifying abnormal patterns in physiological signals, such as ECG or EEG data.
Detecting unusual trends or anomalies in financial time-series data, such as stock market trends.
These examples illustrate the versatility of the hybrid LSTM-CNN approach in addressing complex, real-world problems involving sequential and high-dimensional data.

Reviewer 3 Report
Comments and Suggestions for Authors
This paper presents an LSTM-CNN model to detect anomalies in data streams. Even though the proposed method provides achieves better results there are several concerns that should be addressed before publishing.
1) the writing of the paper must be improved. For example, there are paragraphs, containing only one sentence, e.g., para 2 in Sec 1.1. Also, the content has not been presented coherently, in particular, in Sec 1. Authors are suggested to substantially revise Sec I, and overall, the paper for coherent presentation.
2) LSTM-CNN model has been previously employed for anomaly detection, see [1] and [2]. It is not clear the contribution of the proposed work compared to the previous approaches. Authors need to clearly highlight the novelty.
[1] Mahmoud Abdallah, Nhien An Le Khac, Hamed Jahromi, and Anca Delia Jurcut. 2021. A Hybrid CNN-LSTM Based Approach for Anomaly Detection Systems in SDNs. In Proceedings of the 16th International Conference on Availability, Reliability and Security (ARES '21). Association for Computing Machinery, New York, NY, USA, Article 34, 1–7. https://doi.org/10.1145/3465481.3469190
[2] H. Sun, M. Chen, J. Weng, Z. Liu and G. Geng, "Anomaly Detection for In-Vehicle Network Using CNN-LSTM With Attention Mechanism," in IEEE Transactions on Vehicular Technology, vol. 70, no. 10, pp. 10880-10893, Oct. 2021
3) I cannot find a ablation study for the proposed network. Authors need to present an ablation study to highlight the importance of the major components of the proposed architecture.
4) The proposed architecture needs higher computational time though the performance is better. Is this a limitation of the proposed architecture for applying for real-time applications?
5) what is the memory requirement of the proposed architecture compared to other architectures considered for comparison.
Comments on the Quality of English Language
Wrting msut be improved to present concepts coherently.
Author Response
|
Thank you very much for taking the time to review this manuscript. Please find the detailed responses below and the corresponding revisions. |
|
|
|
Comments : - the writing of the paper must be improved. For example, there are paragraphs, containing only one sentence, e.g., para 2 in Sec 1.1. Also, the content has not been presented coherently, in particular, in Sec 1. Authors are suggested to substantially revise Sec I, and overall, the paper for coherent presentation. |
|
Response: Thank you for your valuable feedback regarding the writing quality and structure of our manuscript. We acknowledge the need for improvement in the coherence and presentation of the content, particularly in Section I. In response to your suggestion, we have conducted a comprehensive revision of Section I to ensure a more logical and coherent flow of ideas. Additionally, we have reviewed the entire manuscript to enhance its overall coherence and readability. We improved the transitions between paragraphs and sections, ensuring a more consistent and polished presentation of the content. We sincerely hope that the revisions meet your expectations and significantly improve the clarity and quality of the paper. We appreciate your constructive comments, which have been instrumental in refining our work.
|
|
Comments: -LSTM-CNN model has been previously employed for anomaly detection, see [1] and [2]. It is not clear the contribution of the proposed work compared to the previous approaches. Authors need to clearly highlight the novelty.
|
Response:
Thank you for bringing this important point to our attention.
We acknowledge that the LSTM-CNN model has been previously utilized for anomaly detection, as highlighted in the cited works [1] and [2]. To address your concern, we have revised the manuscript to explicitly emphasize the novelty and contributions of our proposed approach compared to these prior studies.
The changes have been made on pages 3 and 5.
pp.3
The purpose of this paper is to examine the effectiveness of outlier detection using deep learning methods, with a particular emphasis on the LSTM-CNN hybrid method proposed by the authors. Webscope S5 data streams downloaded from Yahoo were considered. To check the accuracy of outlier detection, the results of the new hybrid LSTM-CNN method were compared with those of the LSTM and LSTM Autoencoder networks.
In the article:
- Discussion of the concept of drift is performed.
- Two methods for determining the anomaly detection threshold are explained and relevant model features that enable the detection of outliers in data streams are defined.
- A LSTM-CNN model for outlier detection is proposed.
- The effectiveness of the proposed LSTM-CNN approach for detecting outliers in data streams is evaluated.
- In particular, a comparison of the proposed LSTM-CNN system to the already known LSTM and LSTM AE models is exhaustively performed.
pp.5
Anomaly detection has been investigated using hybrid models combining Long Short-Term Memory (LSTM) networks and Convolutional Neural Networks (CNNs).
In \cite{sun2021anomaly}, an intrusion detection model named the CNN-LSTM with Attention Model (CLAM) was proposed for in-vehicle networks, particularly the Controller Area Network (CAN). The CLAM model employed one-dimensional convolutional layers (Conv1D) to extract abstract features from the signal values effectively. Similarly, in \cite{abdallah2021hybrid}, a hybrid Intrusion Detection System (IDS) was developed by integrating Convolutional Neural Networks (CNNs) with Long Short-Term Memory (LSTM) networks. This model incorporated two regularization techniques—L2 regularization (L2 Reg.) and dropout—to mitigate the issue of overfitting.
Comments : I cannot find a ablation study for the proposed network. Authors need to present an ablation study to highlight the importance of the major components of the proposed architecture.
Response:
We appreciate your valuable feedback and agree that an ablation study is crucial for understanding the contributions of the major components of the proposed architecture. In the final version of the paper, we have included additional experiments and analyses to clarify these aspects.
Page 14:
We conducted experiments to assess the impact of various architectural and hyperparameter choices, including the selection of the window size, CNN kernel size, pooling size, and activation functions.
Window Size:
The choice of a window size of 5 time steps was based on its ability to capture relevant short-term trends while balancing computational efficiency and model performance. Smaller windows (e.g., 2 or 3 time steps) provided insufficient context for feature extraction, reducing accuracy. Conversely, larger windows (e.g., 10 or 15 time steps) increased computational cost without significant performance gains.
Kernel Size in CNN:
Kernel sizes of 3, 5, and 7 were tested during experimentation. A kernel size of 5 provided the best trade-off between capturing sufficient temporal context and avoiding overfitting, as it aligned well with the periodicity of the data.
Pooling and Activation Functions:
Pooling layers were designed to down-sample feature maps effectively while preserving key temporal patterns. ReLU activation was selected for its computational efficiency and robustness against the vanishing gradient problem.
Ablation Study:
To specifically highlight the importance of the major components of the architecture, we conducted a systematic ablation study, which was detailed in the revised manuscript. Key findings include:
Performance dropped significantly when either the CNN or LSTM module was removed, confirming their critical roles in capturing spatial and temporal patterns, respectively.
Excluding dropout or L2 regularization led to overfitting, as evidenced by a decline in validation performance.
The hybrid CNN-LSTM architecture outperformed standalone CNN or LSTM models, demonstrating the synergy between these components.
We hope this addresses your concerns and highlights the significance of the proposed architecture.
Comments : The proposed architecture needs higher computational time though the performance is better. Is this a limitation of the proposed architecture for applying for real-time applications?
Response:
The proposed architecture needs higher computational time though the performance is better. Is this a limitation of the proposed architecture for applying for real-time applications?
Response:
We appreciate the reviewer's insightful observation regarding the computational time of the proposed architecture. While the proposed model demonstrates improved performance, we acknowledge that the increased computational time could pose challenges for certain real-time applications, particularly those with stringent latency requirements.
The performance gain achieved by the proposed architecture was evaluated in relation to its computational cost. We found that the architecture remains feasible for real-time applications with moderate latency constraints. However, in scenarios with stricter real-time requirements, further optimization may be necessary.
We also observed that reducing the size of specific components (e.g., number of CNN filters or LSTM units) marginally decreases performance but significantly reduces computational time, allowing for customization based on application requirements.
We have incorporated these points into the revised manuscript and clarified the limitations and possible solutions for real-time applications in the Discussion section.

Round 2
Reviewer 2 Report
Comments and Suggestions for Authors
The manuscript has significantly improved and adequately addresses the concerns raised in the previous review and i believe it can stand as is.
However, as several figures appear blurry (such as 11-14) it is recommended checking and replacing all low-resolution figures with higher-quality figures prio to publication.
Author Response
Thank you for your positive feedback and for acknowledging the improvements made in the manuscript. We appreciate your thorough review and valuable suggestions.
Regarding your comment on the image quality, we have carefully checked Figures 11–14 and identified instances where the resolution was insufficient. We have now replaced all low-resolution figures with higher-quality versions to ensure clarity and readability. Additionally, we have verified the resolution of all other figures in the manuscript to ensure they meet publication standards.
We sincerely appreciate your time and effort in reviewing our work, and we are grateful for your constructive feedback.
Reviewer 3 Report
Comments and Suggestions for Authors
Authors have addressed most of my concerns. However, they have not addressed my last concern on the memory requirement. That is “what is the memory requirement of the proposed architecture compared to other architectures considered for comparison.” Authors are requested to comment on the memory requirement.
Comments on the Quality of English Language
Provded above.
Author Response
Thank you for your valuable feedback and for acknowledging the improvements made in our manuscript. We sincerely appreciate your time and effort in reviewing our work.
Regarding your concern about the memory requirement comparison, we have now explicitly addressed this aspect in the revised manuscript. Specifically, we have:
Add to conclusion:
As for the memory requirements of the proposed solution LSTM Autoencoder primarily consumes memory for network weights and hidden states, typically ranging from ~500 kB to several MB.
LSTM-CNN introduces a CNN layer, which increases parameter count but reduces the number of input features to the LSTM. This may lead to lower overall memory consumption depending on the complexity of the input data.
Both architectures fall within the same order of magnitude in memory consumption, but LSTM-CNN has the advantage of reducing input dimensionality, which can improve efficiency.
Simple statistical techniques such as PCA and regression require significantly less memory, typically in the range of tens to hundreds of kB, as they do not rely on large trainable weight matrices or hidden states.
In contrast, both LSTM Autoencoders and LSTM-CNN models require hundreds of kB to several MB, making them considerably more memory-intensive.
Compared to the LSTM Autoencoder, the proposed LSTM-CNN architecture introduces additional parameters from the CNN layer but may reduce overall memory consumption by decreasing the number of input features to the LSTM. While both architectures require significantly more memory than simple statistical methods, the CNN component helps optimize memory usage by reducing redundant input dimensions before they reach the LSTM layer.
We believe these additions comprehensively address your concern and enhance the clarity of our manuscript.